
# PREGRIDBAL 1.0: towards a high resolution rainfall atlas for the Balearic Islands (1950-2009)

Toni López Mayol[1], Víctor Homar[1], Climent Ramis[1], José Antonio Guijarro[2]

[1]Grup de Meteorologia. Departament de Física. Universitat de les Illes Balears. 07122 Palma (Mallorca)
[2]Delegació Territorial en Illes Balears. Agencia Estatal de Meteorologia. Moll de Porto Pí, s/n. 07015 Palma (Mallorca)

*Correspondence to*: Victor Homar (victor.homar@uib.cat)

**Abstract.** This work presents a catalog of daily precipitation in the Balearic Islands created with data from AEMET assistant observers, including registers since 1912. The original digital daily data file has been interpolated onto a regular 100m
resolution grid (namely PREGRIDBAL), defined with the aim of becoming a valid standard for future methodological improvements and catalog upgrades. Daily precipitation amounts on each grid-point are calculated using an analysis method based on ordinary Kriging, using the daily anomaly with respect to the annual mean for all available observations each day. Due to quality concerns, the time span for products derived from the catalogue is limited to the 1950-2009 period. Therefore, from the elementary daily maps, monthly-, annual-, quinquennial-, and decadal-accumulations are produced. Similarly, the
catalog allowed for quantification of climate trends in rainfall amounts in the Balearic Islands, with the significant advantage of minimising the biases originated from heterogeneities in the spatial distribution of stations across the archipelago.

Results show a general decrease in precipitation during the 1950-2009 period. From 1950 to 1979, the average annual precipitation across the islands was 624,3 mm, while from 1980 to 2009 diminished to 555,36 mm. Changes in precipitation
patterns, which vary among the different areas, are also detected. Most significant reductions (over 80% significance on the trend) are found in the north half of the archipelago and especially in Mallorca, where the Tramuntana mountain range stands out. All seasonal trends show a decrease, with values ranging between 1 and 3mm per decade, with the exception of autumn, which reaches up to 7mm per decade. October shows the most dramatic decrease (-10,34 mm per decade) and, conversely, September and November show an increase in precipitation (3,28 and 1,82 mm per decade, respectively) with a statistical
significance above 85% in almost all the archipelago, and even exceeding 95% in the Pitiüses.

## 1 Introduction

The Mediterranean region has been deemed as 'especially sensitive' to the impact of climate change by the International Panel on Climate Change (IPCC, 2007). The Balearic Islands show notable variability in precipitation trends during the 20[th] century, and changes are lower than 3% for the last part of the 20[th] century (Homar et al., 2010). Global warming is causing
a higher divergence of water vapour, which will translate into a decrease of precipitation and a possible higher number of dry



years in the future, with this trend generalising to the subtropical areas (Clivar España, 2010). The Iberian Peninsula does not show significant decreasing trends and the lack of significance is linked to the short length of the time series before 1980. However, the precipitation data over the last decade show a clear significant negative trend. In contrast to statistically better behaved atmospheric magnitudes, precipitation inevitably requires more complex analytical methodologies. On the one hand, precipitation shows a strong spatial and temporal variability in all scales of interest. On the other hand, most precipitation climatic series available nowadays show temporal gaps longer than their autocorrelation time, thus extremely difficulting a thorough and exact analysis.

There have been multiple initiatives to generate homogeneous grid-precipitation catalogues for sufficiently long periods so that solid and significant climate analyses are viable. At the global scale, CRU database (Climate Research Unit, University of East Anglia, http://www.cru.uea.ac.uk) focuses on monthly accumulation of precipitation on a 2.5ºx3.75º grid (lat x lon). On a continental scale, Haylock et al. (2008) created the E-OBS database, which includes a 50km gridded analysis of daily precipitation over Europe. In Spain, Herrera et al. (2012) developed a 0.2º grid which covers the mainland and the Balearic Islands for the 1950-2003 period. This database, named *Spain02*, allows for the analysis of the regional evolution of daily precipitation for scales larger than 20km, but it does not allow to resolve local structures in areas with high geographic variability, such as the Balearic Islands, where the density of observatories contributing precipitation data to the State Meteorological Agency (AEMET) offers a more accurate analysis. In that sense, Homar et al. (2010) found significant reductions in precipitation using data from 20 stations in the Balearic Islands, with an average decrease in annual precipitation of 16.3mm per decade (mm/dec) during the period 1951-2006. However, the representativeness of their findings can be questioned taking into account that 8 out of the 20 stations included in the study are located in the western half of the Tramuntana mountain range, and they use no stations in the north quadrant of Mallorca, nor in Eivissa, since no station in the database meets the representativeness requirements imposed by the authors.

Guijarro (1986) defines a grid over the Balearic Islands with a 1km resolution and generates average precipitation fields for the period 1961-1980, covering the four main islands. The results of his multiparametric interpolation (including height, sea distance, terrain gradient,…) show the climatic diversity of the Balearic Islands. On one side, the Tramuntana mountain range shows average annual precipitations over 1000 mm, with peaks higher than 1400 mm. Guijarro's results (1986) estimate that in central and eastern parts of Mallorca 500-700 mm are collected, and on the meridional coast barely reaches 300 mm per year. Menorca shows a remarkable uniformity with more than 600 mm annual average, and peaks over 700 mm. However, in Eivissa the contrast between the interior –more than 600 mm annually- and the coast –slightly over 350 mm- reaches up to 250 mm in less than 10 km.

In the current context of assessing climate change impact on the availability of natural resources, and considering the existing results, which indicate great variability in precipitation in the Balearic Islands, the need for a detailed analysis of



local characteristics and precipitation evolution in the last decades becomes especially relevant. This article proposes a definition of a high-resolution geospatial grid with the intention of becoming a standard for the study of precipitation in the Balearic Islands for the coming years. Furthermore, it presents the first catalogue of daily precipitation maps on this reference grid for the period 1914-2009, generated on a geostatistical model based on ordinary kriging (Matheron, 1963).

The ultimate goal and subjacent motivation is the analysis of precipitation trends in the Balearic Islands in the last decades, accounting for the entire archipelago's territory and avoiding biases derived from the inhomogeneity in the spatial distribution of historical series currently available.

We define PREGRIDBAL grid with the aim of becoming a long-lasting reference for the characterisation of precipitation in
the Balearic Islands. Considering the effort made to coordinate different nets of atmospheric observation in the Balearic Islands, with the relatively recent installation of the AEMET radar, continuous improvements in precipitation estimation techniques using satellite measurements, and due to the large geographical variability of the archipelago, it has been decided to define a squared grid with a homogeneous spatial resolution of 100 m. This resolution is clearly higher than the spatial scales resolved by the current AEMET climatic databased used in this study. The reason for this apparently unnecessarily
high resolution is to facilitate the characterisation and assessment of future improvements to the precipitation catalogue, derived both from methodological improvements and from the addition of new sources of information for the analysis.

The first part of the article discusses quality and characteristics of daily precipitation series from the AEMET assistant observers station network in the Balearic Islands, and includes a detailed presentation of the geostatistical modelling
methodology used to generate the daily precipitation catalog. Secondly, results of this interpolation are shown through the analysis of 2 historic episodes of rain in Mallorca. To conclude, some products derived from the catalogue are presented, especially the trends observed in annual precipitation in the Balearic Islands. The last section summaries the study's most important achievements and opens new lines of work for the future.

**2 Data and methodology**

**2.1 Data compilation**

Data used in the study come from pluviometric registers from the AEMET assistant observers' stations network database. It contains registers of daily precipitation (from 07 UTC day D to 07 UTC day D+1) in the Balearic Islands from 418 stations since 1912 (Fig. 1). In this study all available data from 1914 to 2009 have been used, applying a basic quality control so as
to eliminate non-physical precipitation values. For the first two years, the grid consisted of 3 (1912) and 2 (1913) stations, and it was not until 1914 when data from 4 stations were first collected. The daily map catalogue on PREGRIDBAL has





been generated from 1914 onwards. Average spatial density of stations on the grid, leaving aside asynchrony in the running of many of them, is 0,08 stations per km$^2$ (or 3,5 km average distance between neighbouring stations).

## 2.2 Methodology

Data availability in the base shows how during the 1950-1970 decades the network of stations experienced a remarkable
increase both in number of operational stations and in the number of measurements collected (fig. 2). During the last decade, the number of operational stations reached 209, even though there is a significant reduction in the number of measurements in the last years attributed to problems in recording and digitalising data. It is worth noting that changes in location of stations in this database lead to defining a new ID for the station, so from the analysis perspective they are the equivalent to the disappearance and installation of independent stations. In our calculations, days with no record from a station, are treated
as if the station was not present.

From a quality-control perspective of original precipitation data, daily analyses become a complementary testing element which allows for assessing the spatial consistency and coherence of observed amounts through their influence on the final analysed field, which can be used in the future so as to advance in the depuration of original measurements.

## 2.3 Analysis

Primary products conforming the high-resolution daily precipitation catalogue in the Balearic Islands have been generated on a regular grid. The PREGRIDBAL grid is thus defined as land points every 100 m within the boundaries set by x=345000 and x=614000, y=4278000 and y=4438000, in UTM coordinates (zone 31).
On the PREDGRIDBAL grid, we generate a daily precipitation analysis for every day for which we have data. We use
ordinary kriging (Matheron, 1963) to calculate daily precipitation values on the grid's nodes starting with registers available for a given day in the AEMET observers' database, thus solving the estimation uncertainty (Dingman et al. 1988): an improved technique after Krige's work (1951) and which has recently advanced drastically in its application in the climatology field and its respective variables (Moral, 2009).

Variogram analysis (variance vs distance between stations in the database) suggests the convenience of defining an exponential model of variogram so as to obtain the best adjustments to the sample variance. Therefore, for every day in the database, an exponential variogram has been adjusted to a variogram of daily precipitation values for the day (with a location limit of 50 km) and this has been used in the krigging algorithm. This is how the analysis is adapted (through the adjusted variogram parameters) to the spatial characteristics of the precipitation field for the day under analysis, with spatial
correlation scales of the adjusted field for days of very local convective rain, for example, being different from those days of general stratiform rain over all the territory.



Due to a highly biased distribution towards low values (i.e. gamma distribution), and the strong spatial variability shown by daily precipitation values, we do not analyse absolute measured values of daily precipitation ($p_d$) but the anomaly with respect to average annual precipitation ($p_a$). In particular, first for every station in the database we calculate $p_a$ and analyse those precipitations on the PREGRIDBAL grid. This gridded field ($p_{ag}$) is static and a reference for the catalogue. Then, for every day we apply the krigging algorithm using as predictive variable the daily precipitation anomaly in each station with respect to $p_a$: $p_a^r = p_d/p_a$. Finally, the day's absolute daily precipitation field on the grid is obtained by multiplying the resultant krigging estimation by the reference annual average precipitation field ($p_{ag}$) mentioned before.

This process is applied to every day from January 1, 1914 to December 31, 2009.

### 2.3.1 Instrumental Analysis

Pluviometric daily precipitation measures archived at the AEMET observers network  database are not free of systematic errors, which must be taken into account and incorporated to geostatistical calculations so as to quantify error associated to precipitation estimations obtained by applying the krigging method. In the case of manual pluviometer readings, the main sources of errors are calibration and precision of graded scale, human reading errors and errors originated in the relationship between the pluviometer's shape and wind characteristics during reading period. Barcenas (2012) estimates a typical error of up to 5% in standard pluviometers through the comparison of a long series of readings of multiple pluviometers set together. For all purposes, a 5% error will be attributed to daily precipitation values in the original data, which will be transferred to resulting analysis fields.

### 2.4 Tendency calculation

Once daily precipitation maps and the derived accumulations (monthly, seasonal, annual…) have been generated, temporal series for every gridpoint are available. Naturally, the catalog allows for calculating temporal tendencies and opens the opportunity for their spatially homogeneous representation on the Balearic Islands. In this study, lineal tendencies have been calculated with an algorithm based on the MM estimator proposed by Yohai (1987) and an efficient iterative procedure to recalculate least-squares weights (Jennrich and Moore, 1975), using Rousseeuw's implementation (Rousseeuw et al. 2008). Standard errors in calculated tendencies and confidence intervals were obtained following the procedure established by Crouch et al. (2003). Even though there are techniques for the regional analysis of tendencies in the assessment of supraregional forcing impact (e.g. Renard et al. 2008), this study focuses on local changes given the geographical diversity of the Balearic Islands.



### 3 Products

#### 3.1 Daily precipitation catalogue

In order to illustrate the generated daily products, we present two relevant pluviometric episodes in the Balearic Islands. First, an episode which took place on September 6 1989, characterized by a cold intrusion at mid levels, which led to the

generation of a convective complex (Capel, 1989; Quereda i Obiol, 1990 ) affecting, among other places, the southwest area of the archipelago, and which caused important precipitations in Mallorca: Picot (250mm), Cala Ratjada (198mm) or Port of Manacor (196mm). The resulting analysis (Fig. 3) accurately represents those values and shows highly realistic structures due to two factors: the high density of available stations on that day; and also the usage of precipitation ratios during the process of analysis using krigging method, which adds information onto the resulting field about the spatial correlation of

precipitation through the annual average field ($p_a$).  It is worth noting that, even though the general structure of the analysed field is realistic, some stations clearly show values which are significantly different from those in the surrounding areas, thus indicating archived values of precipitation which probably require a more refined quality control. The availability of daily analysis presented in this study will help in future control tasks of the original AEMET database.

A second relevant episode took place on November 11 2001, affecting most of the islander territories (Fig. 4) and especially

the Tramuntana mountain range, where daily accumulations of 220 mm were measured. This case is especially relevant to verify the skill of the designed analysis method to combine heavy stratiform rain across the territory with isolated cells of intense convective precipitation.

#### 3.2 Monthly, seasonal and annual accumulations

Starting from the historical catalogue of daily precipitation, different fields of accumulation –monthly, seasonal, annual, quinquennial, and decadal- over 30 years are obtained. Evolution in the size of the network of stations (Fig.2) shows the increase in the number of operational stations from the 1950s onward, reaching the 100 units (equivalent to an average station-station distance of 7 km). As a consequence, we limit the calculation of all derived products to the 1950-2009 period. As an example, Fig. 5a shows accumulated precipitation in 1972, the year with highest accumulated annual precipitation in

the Balearic Islands since 1950, with values higher than 2000 mm in the Tramuntana mountain range, and with anomalies with respect to the 1950-2009 average of +191 mm (Fig. 5b).

#### 3.3 Average values in the Balearic Islands

Table 1 shows average precipitation values as for years, months and stations in the Balearic Islands, obtained by spatial

averaging the accumulation fields described in the previous section. A novelty of this work as compared to previous ones (e.g. Homar et al. 2010), these spatial averages have significantly reduced de distortion and biases caused by the spatial





heterogeneity of the network of stations. Naturally, most precipitations in the Balearic Islands concentrate on autumn (September, October, November), and contrarily summer months (June, July and August) conform the dry season. July and October are, respectively, the driest (7,43 mm) and wettest (96,57 mm) months in the archipelago. According to our convention, December, January and February represent winter, while March, April and May conform spring.  Average

annual precipitation on the archipelago, calculated on the PREGRIDBAL grid, shows a highly notable interannual variability with a long-term oscillation of precipitation with an average period of between 17 and 20 years (Figure 7).

## 4 Tendencies observed

### 4.1 Annual tendencies

On one side, the least-squares fit to the annual precipitation series averaged across the Balearic Islands shows a negative tendency of -12,76 mm/dec for the period 1950-2009 (Fig. 7), with a confidence level of 73%, below the conventional 95% threshold. Confidence level indicates the probability of the process or system that generates the data (i.e. climate in the Balearic Islands in this analysis) changing, as opposed to the situation in which the climate remains stationary and, the series shows a non-zero tendency (random), compatible with the system stationary variability. The highest intrinsic variability a

series shows (e.g. annual precipitation), the most probable the resulting tendency is a random result and not a tangible change in the climatic system under analysis. If we compare Homar et al. (2010) estimation for the 1951-2006 period (based on 20 stations), which shows a negative tendency of -16,33 mm/dec (with a confidence degree over 80%), with the tendency calculated for the same period on the PREGRIDBAL grid, of 20,63 mm/dec, it becomes evident that the representativeness bias has a strong impact on the Homar et al. (2010) analysis and results. De Luis et al. (2009) calculate precipitation

tendencies on the Spanish Mediterranean basins (excluding the Balearic Islands) for the period 1951-2000 and they obtain a decrease of up to 13,9 mm/dec for annual precipitation, which is perfectly consistent with the spatially homogenized results on the PREGRIDBAL grid.

On the other side, the availabity of high resolution precipitation fields allows for a detailed analysis of the spatial distribution

of precipitation tendencies across the domain. Therefore, at each gridpoint we calculate a linear tendency, from the 60 annual accumulation values (1950-2009). The resulting tendency field (Fig. 8) shows that in general, precipitation has decreased in the northern half of the islands, especially on the Tramuntana mountain range in Mallorca and also remarkably to the north of Menorca, with confidence levels for the negative sign of the trend between 70% and 90%, and 95% respectively (Fig. 9). These reductions oscillate between 20 and 40 mm/dec. Eastern Mallorca is also representative, with Artà and Manacor areas

showing results of -20 to -25 mm/dec, with little statistical significance. Areas close to Muro or Sa Pobla also show a decrease in precipitation with an average rhythm of 20 mm/dec, whereas Bunyola, Alaró, Soller and surrounding areas are




far more noticeable, with trends reaching -35 mm/dec. Even so, the area where pluviometric decline impacts the most is the north of Menorca and the municipality of Pollença, north of Mallorca.

Areas showing more stability and therefore non-existent or non-important variations are located on the Mallorcan southwest and south, near Palma beach. A possible explanation is related to the increase in urban areas nodes, associated to the higher presence of pollutant particles from the city and the suburbs. Nevertheless, statistical confidence for these reductions is especially low, around 5 to 40% as a general trend in the south half of Mallorca, the southwest coast of Eïvissa and Menorca, where we find the regions with less noticeable tendencies; reductions smaller than -5 mm/dec. Conversely, in Menorca, confidence levels are higher than 80% in all its territory, except for a small area close to Ciutadella. In Eïvissa variations are more irregular, being higher on the east coast and lower on the west and to the north.

### 4.2 Seasonal tendencies

As for seasonal attribution of changes observed in annual accumulation, winters do not show great variations, with a seasonal tendency of -2,15 mm/dec. A higher decrease is found on the northwest and northeast of Mallorca, with values ranging between -5 and -25 mm/dec, and a degree of statistical confidence reaching 95% in areas of Porto Cristo, Port de Pollença, Bunyola and Esporles. The Tramuntana mountain range is especially relevant, and also, but with an inferior decline, the east of Mallorca, where, with the exception of places previously mentioned, confidence levels oscillate between 50% and 99%. The case of Pollença is especially remarkable, with a decrease of -25 mm/dec. Menorca shows increasing reductions from west to east, with maximum values of -15 to -20 mm/dec in the north of Maó, with a confidence level of 85-95% in the north coast and the eastern area, and 50-60% in the rest. In winter, December shows a mild positive tendency with little variability during the last 60 years (0,2 mm/dec), whereas January, as a whole, decreases in -3,99 mm/dec, presenting a homogeneous decreasing pattern or around - 5 to -7 mm /dec in all the territory, being more negative in the Tramuntana mountain range.

Changes in spring are similar to changes in winter: maximum decreasing values are less important but regions with lower tendencies are widening. In fact, spring in the Balearic Islands presents a seasonal tendency of -2,36 mm/dec, which is similar to winters. Eïvissa shows a remarkable decrease of around -10 to -15 mm/dec, especially around Santa Eulàlia and Es Canà, with a confidence level exceeding 95%. In Mallorca reductions are concentrated on the Tramuntana mountain range, with tendencies in the interior of the island no higher than -5 mm/dec (Algaida). Eastern Mallorca shows a decrease of -12 mm/dec in Capdepera and the coast of Manacor. Menorca does not show any significant variations except for the northeast coast, where the decrease ranges from -4 to -7 mm/dec, with a low degree of confidence. March is the month showing the largest decrease, with -4,27 mm/dec, whereas May shows an increase of 2,61 mm/dec.





Summers show no reductions larger than -20mm/dec anywhere in the Balearic territory, although in areas of the municipality of Manacor and the northwest corner of the Tramuntana mountain range exceed the -15 mm/dec level. However, the average seasonal tendency in all the territory in summer is -1,62 mm/dec and the confidence level of the trend is higher than 95% in most of the archipelago, especially in the eastern part. Moreover, in Menorca, with the exception of the surroundings of

5 Ciutadella, it is higher than 95%. Mallorca, on the other hand, presents an eastern half with over 95%, whereas areas in Palma bay, Andratx and Inca show very low confidence levels. The Tramuntana mountain range also shows confidence levels higher than 85%, especially its northern section. Menorca and the Pitiüses show reductions between -5 and -15 mm/dec, whereas half the Mallorcan territory, including Andratx, Palma, Llucmajor, Campos and most of the interior show barely significant tendencies. It is worth noting that Escorca shows the most dramatic reduction (-17 mm/dec) across the

10 Balearics. During the month of June there is a generalisation in reductions of -2 mm/dec (Llucmajor, Palma, Bunyola, Valldemossa, Soller, Fornalutx, Llubí and Alcúdia in Mallorca; Ciutadella in Menorca and very specific locations in the Pitiüses). August shows modest reductions in general, excep for Palma bay, where values are negligible, being more negative in eastern Mallorca.

It is in autumn when more relevant tendencies are found since it is the only season including months with positive tendencies, achieving the 20 to 30 mm/dec in different parts of the territory (Fig. 11), but with an average seasonal tendency of -7,25 mm/dec. Some notable examples are Andratx, Estellencs, Banyalbufar and Valldemossa, with 30 mm/dec, whereas the mountains show positive tendencies of around 10-15 mm/dec, but with a very low statistical confidence. Another remarkable area regarding pluviometric increase is Palma bay, where even though low values and thus with a low confidence

degree, increase ranges from 5 to 10 mm/dec. Reductions can also be found in the north of Menorca (-25 mm/dec) which moderate when moving to the south of the island, with a decrease of -10 or -5 mm/dec. Whilst in Formentera there is no relevant tendency, Eïvissa shows values of -5 to -15 mm/dec in the centre of the island and part of the north coast, associated with low confidence levels, being inferior in Eïvissa (<60%). October highlights as the month that contribute with  larger reductions to the annual pluviomètric decreases (-10,34 mm/dec), whereas September and November keep a positive

tendency.

These seasonal tendencies can be compared to those obtained by De Luis et al. (2009). For example, precipitation tendency for the Spanish Mediterranean basin in winter resulted in -2,2 mm/dec, identical to that obtained over the PREGRIDBAL grid. It is especially noticeable the difference in autumn tendencies (-1,8 in the basin), possibly caused by the strong regional

variability, a consequence of the dominant convective systems in autumn precipitation in the Spanish Mediterranean.

As for monthly variations, November shows remarkable tendencies, with Mallorca and Menorca showing an important increase in precipitation (Fig. 10), with a maximum tendency of 12 mm/dec in Colònia de Sant Pere. In addition, all the Mallorcan territory shows increases which range from 2 mm/dec in the south to a more generalised 7 mm/dec. Minorcan



tendencies reach up to 8 mm/dec south of Maó, whereas in the interior of Eïvissa and western Formentera the decreases are smaller than -5 mm/dec.

**5 Conclusions**

Using the daily precipitation records included in the AEMET Climatological National Database, a daily precipitation atlas has been created on a regular grid (PREGRIDBAL) covering the territory of the Balearic Islands with a 100 m resolution. The precipitation atlas thus generated contributes to the creation of the foundations which will allow for the study of climatic characterisation of precipitation and its impact on natural and socio-economic systems with a high spatial and temporal resolution covering the Balearic Islands. The method of analysis designed in this first version uses the ratio of precipitation

to average annual precipitation as a predictor variable and an adjusted variogram for each day in the catalogue, thus obtaining better results in a region of great precipitation variability, due to the convective character of an important proportion of the annual precipitation. One of the main motivations for the generation of this catalogue is the analysis of the evolution of the precipitation in the Balearic Islands during the last six decades, with the aim of improving the results obtained in previous studies, which included a severe limiting number of stations, all with a heterogeneous distribution.

Precipitation tendency in the Balearic Islands is clearly negative (-12,76 mm/dec with a level of statistical confidence of 73%), even though it presents zonal variations and clear geographical influences. The archipelago presents a latitudinal variation of precipitation and its tendencies, using as extremes the south of Eïvissa and Formentera (-10 mm/dec) and the north of Menorca (-32 mm/dec). The island of Mallorca presents tendencies with the southern half being closer to Eïvissa's

and the north half showing more similarities to the variations obtained in Menorca.

The end of the 20th century and the beginning of the 21st (last 30 years) registered a 13% less precipitation than the mid 20[th] century, when average annual precipitation (considering the islands jointly) presents a moderate negative tendency (-12,76

25  mm/dec). This decrease is remarkable in Mallorca, where the contrast between the north half (from -15 to -42 mm/dec) and the south half (0 to -15 mm/dec) is very evident. The southern part of Mallorca presents areas with no clear tendencies and even with increases near the city of Palma. This positive tendency could be attributed to the effect of friction which the city has on athmospheric currents or also to the pollution generated in the urban area. Moreover, the north-northwest of the Tramuntana mountain range, and especially the municipalities of Pollença, Soller, Alaró or Bunyola, show remarkable

decreases of -35 to -45 mm/dec over the 1950-2009 period. In Eïvissa, specifically in the area of Santa Eulàlia, the decline of precipitation do not surpass the -20 mm/dec, whereas in Menorca losses are more notable, especially in the northeast part of the island.




As for variations in seasonal accumulation, winter, spring and summer show mild and uniform contrasts, presenting negative tendencies of between -10 to -15 mm/dec, especially to the north of the islands, whereas autumn shows an important increase in the Tramuntana mountain range (up to 25 mm/dec in Valldemossa and Andratx) and milder in Palma and surroundings,

and in parts of Llevant Natural Parc.

Both September and November show a remarkable increase in precipitations during the summer period. September presents a stronger tendency with more important values emerging to the north of the Pitiüses and Menorca, and especially in the Tramuntana mountain range. However, November shows a notable increase in all the territory (especially in eastern

Mallorca), except for the Pitiüses.

A unique specific conclusion which can be derived from the use of the PREGRIDBAL atlas, by cause of covering all the territory of the Balearic Islands, is the identification of the Tramuntana mountain range as a "hot spot" in the archipelago, where the most extreme and significat pluviometric variations are registered, partly attributable to the fact that this zone

registers the maximum annual precipitation values across the islands.

In the future we hope to incorporate improvements to the catalogue both as for data sources used (automatic stations, radar and satellite) and as for analytic techniques, such as incorporating variables like terrain slope and vertical ascents forced by orography, among others.

The resultant atlas is public and can be accessed at http://pregridbal-V1.uib.es

**Acknowledgements**

Data have been provided by the Territorial Delegation of AEMET in the Balearic Island. The study has partially been funded by the *Conselleria d'Educació, Cultura i Universitats del Govern de les Illes Balears* and FEDER fund (project 7/2011 competitive research groups) and Ministry of Science and Innovation (project CGL2011-24458). *Conselleria de Medi Ambient del Govern de les Illes Balears* has also funded the essay through PREGRIDBAL contract.





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





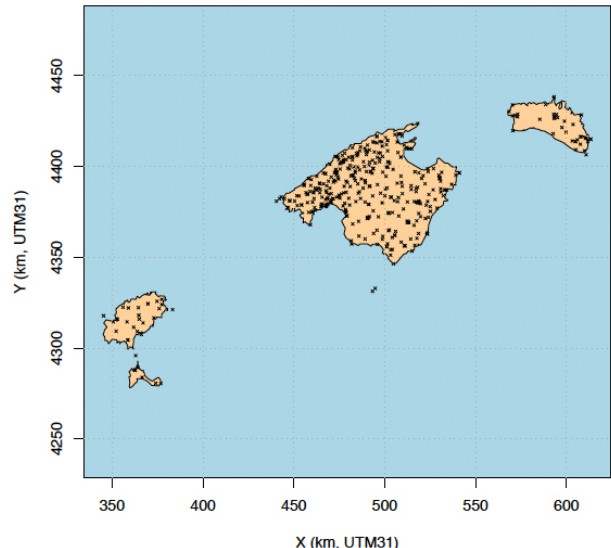

**Figure 1. Location of the 418 rain gauge stations of the AEMET at the Balearic Islands since 1914. There are two stations in Cabrera.**





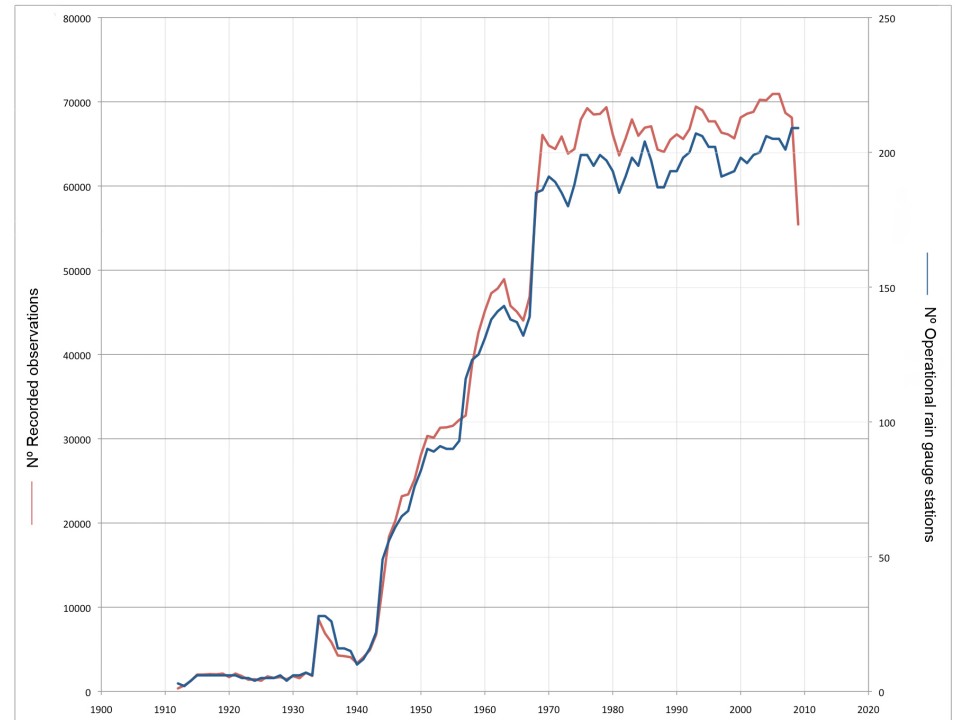

**Figure 2. Evolution of: (Red) Total number of annual rainfall records. (Blue) Number of operational rain gauge stations per year in the network.**





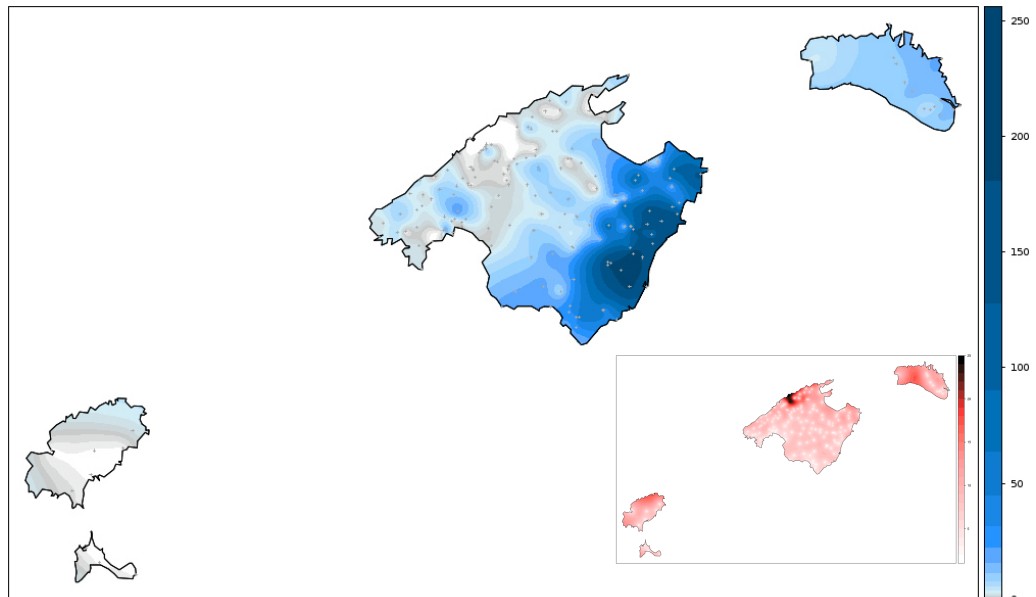

**Figure 3. Map of the daily precipitation (mm) on the 6 September, 1989. The grey dots indicate the position of the stations which were available that day. The inset map shows the field variance resulting from the analysis process.**




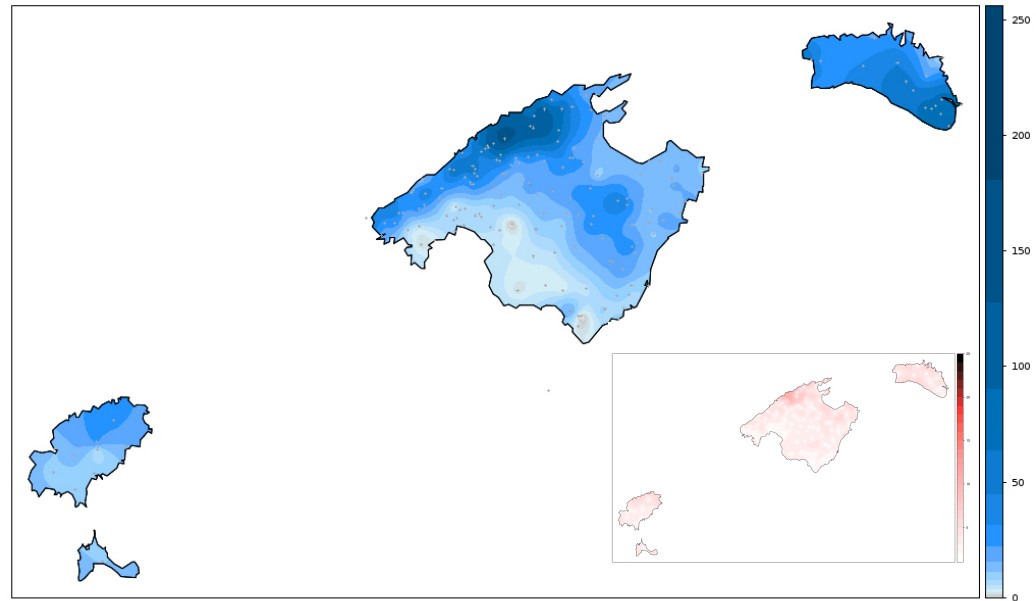

**Figure 4. Similar to figure 3, but corresponding to the data obtained on 11 November 2001. The inset map shows the field variance resulting from the analysis process.**





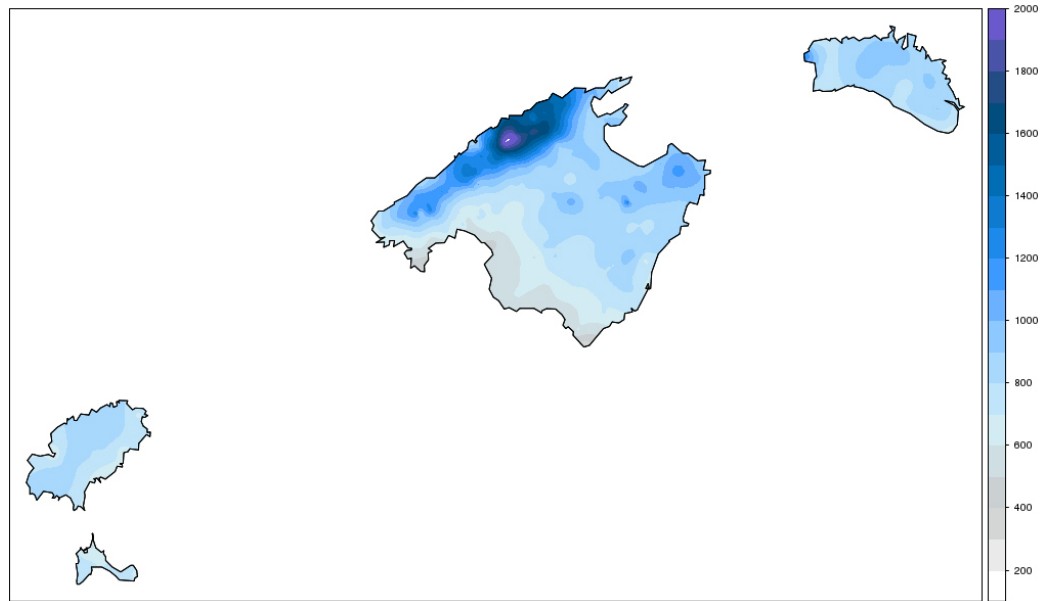

**Figure 5a. Annual precipitation (mm) of 1972.**




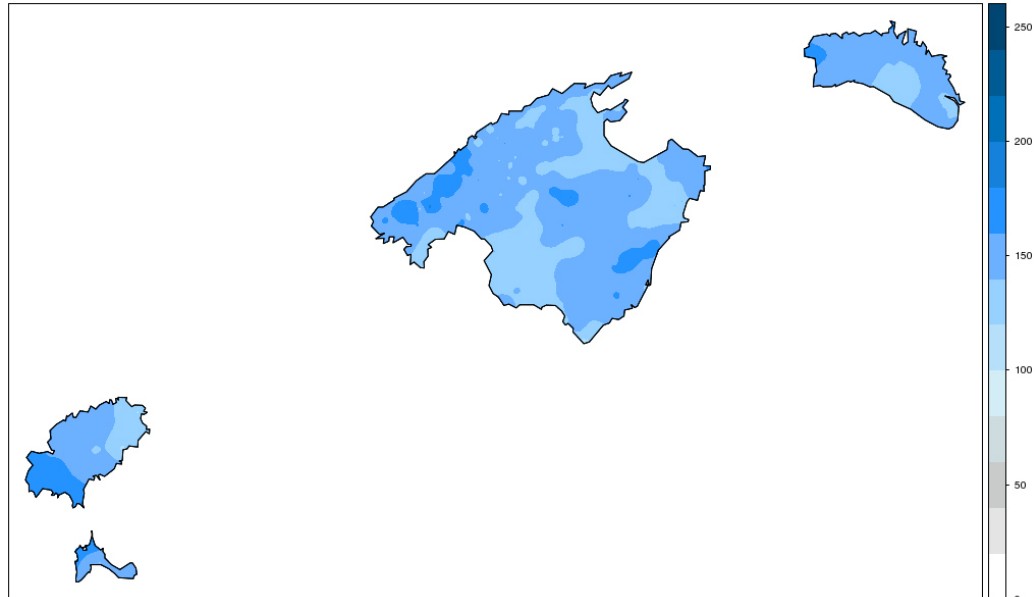

**Figure 5b. Deviation of the annual precipitation (mm) in 1972 regarding the average from 1950-2009.**



| Period 1950-2009 | | | |
|---|---|---|---|
| | | Error | % |
| | Average rainfall (mm) | Variance (mm) | ( |
| **Annual** | 589,8 | 25,2 | 4,3 |
| **Seasonal** | | | |
| Winter (*) | 175,3 | 6,9 | 3,9 |
| Spring | 127,7 | 5,2 | 4,01 |
| Summer | 48,9 | 2,7 | 5,5 |
| Autumn | 237,9 | 9,6 | 2,9 |
| **Monthly** | | | |
| January | 58,1 | 2,3 | 4 |
| February | 45 | 1,8 | 4 |
| March | 45,3 | 1,8 | 4 |
| April | 47,5 | 1,9 | 4 |
| May | 35 | 1,5 | 4,3 |
| June | 17,9 | 1 | 5,6 |
| July | 7,3 | 0,4 | 5,5 |
| August | 23,7 | 1,3 | 5,5 |
| September | 63,1 | 2,9 | 4,6 |
| October | 94,8 | 3,9 | 4,1 |
| November | 80 | 2,9 | 3,6 |
| December | 72,3 | 2,8 | 3,9 |

**Table 1. Average annual, seasonal and monthly precipitations of the Balearic Islands.**

**\*Winter of 1950, including December 1949.**



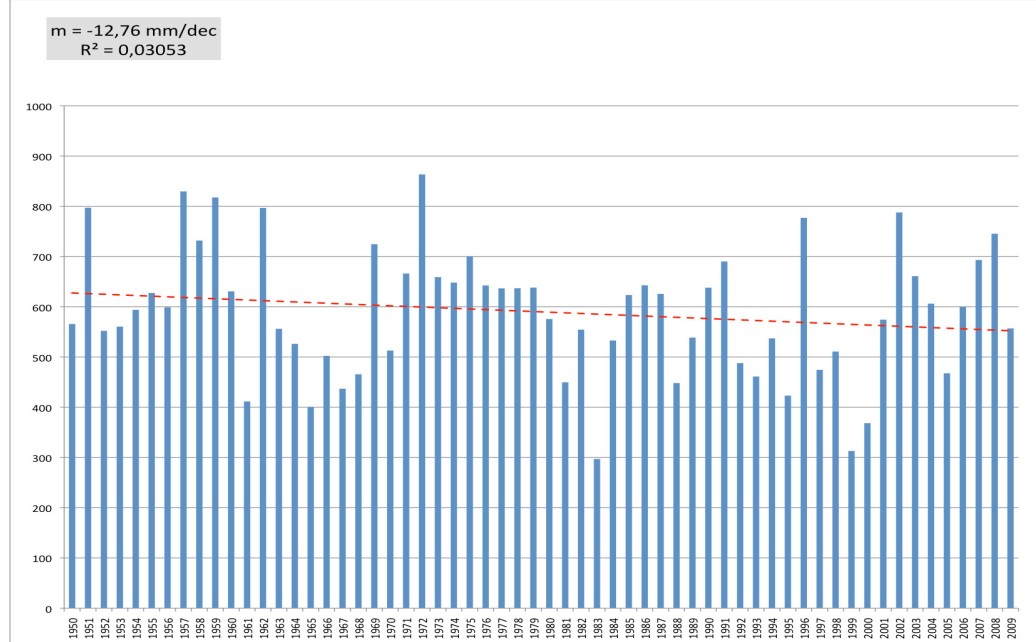

**Figure 7. Average annual precipitations (mm) of the Balearic Islands. (Red) Least Squares fit trend line.**





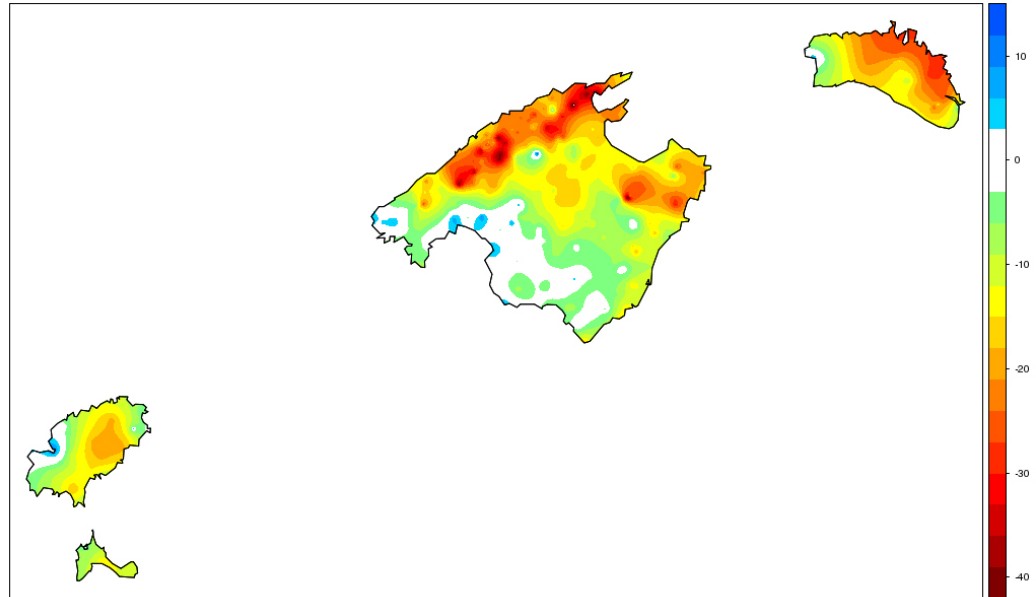

**Figure 8. Trends of the average annual precipitations (mm/dec) across the Balearic Islands for 1950-2009.**




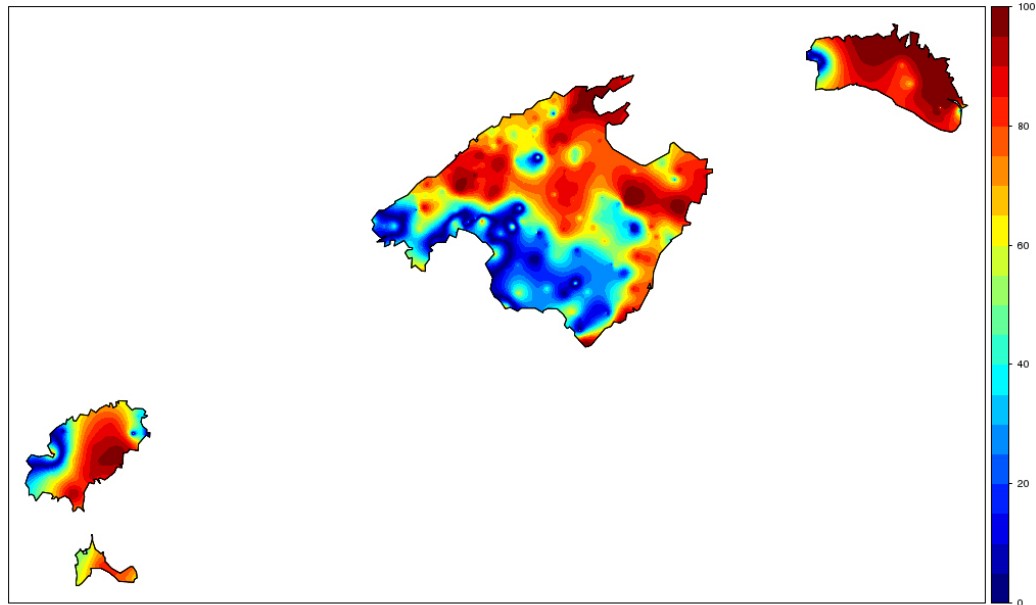

**Figure 9. Level of statistical confidence (%) of the average annual precipitation trend for 1950-2009.**

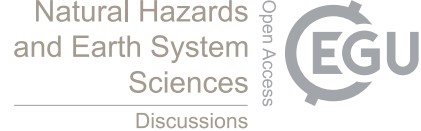

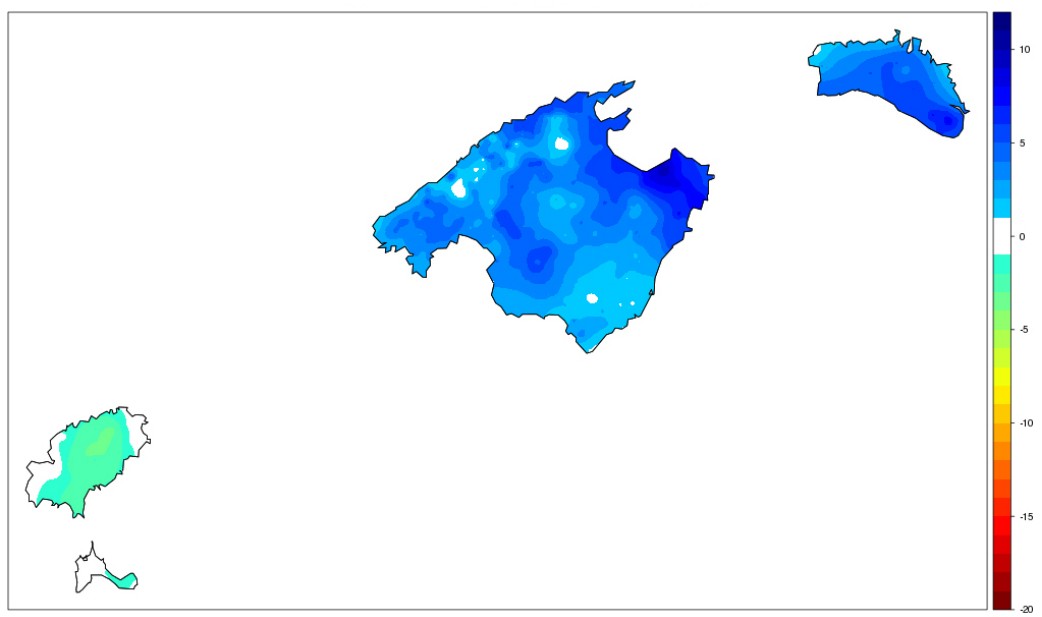

**Figure 10. Trend of the precipitations (mm/dec) in November.**

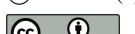



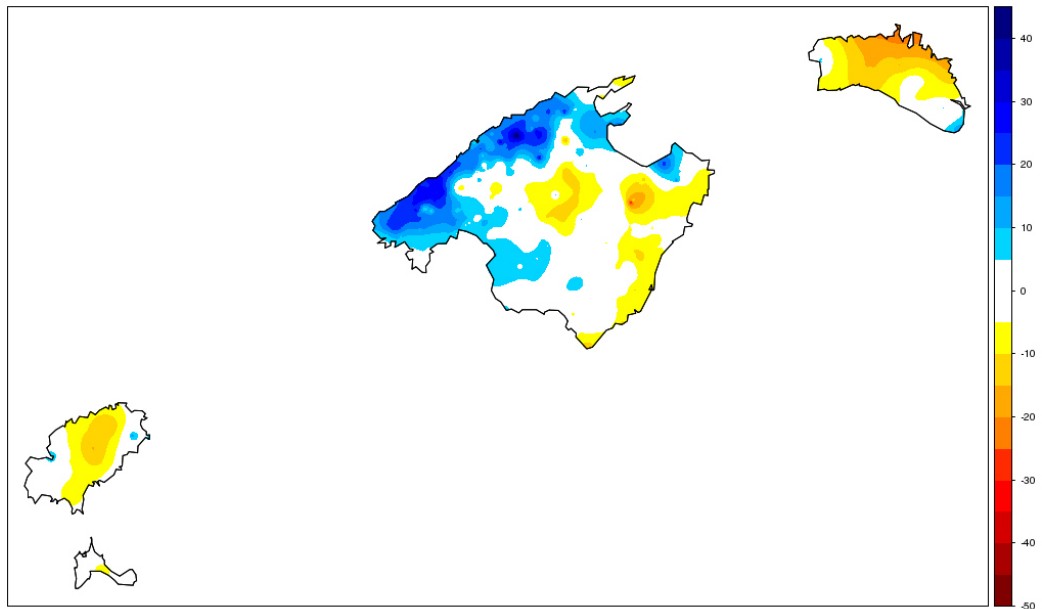

**Figure 11. Trend of the precipitations (mm/dec) in autumn.**