# Peer review of "PREGRIDBAL 1.0: towards a high-resolution rainfall atlas for the Balearic Islands (1950-2009)"

_Natural Hazards and Earth System Sciences, 2016_

## Referee Comment (RC1) · Anonymous Referee #1 · 15 Dec 2016

The topic of this paper is interesting and the new product presented is important, however there are some important issues to be addressed before it can be considered for publication.

One of my main concern is that in the paper the authors do not provide a satisfactory quality analysis of the data. In fact, the authors just write that "In this study all available data from 1914 to 2009 have been used, applying a basic quality control so as to eliminate non-physical precipitation values". What kind of analysis has been applied? Quality control and homogenization of the climate series are necessary to guarantee a reliable analysis. The authors must solve homogeneity problems with the application of a homogenization procedure before the spatial interpolation of the data.

At present, among the various methods to solve homogeneity problems, there is not a single objective one, and the choice of the most suitable procedure is strictly related to the dataset characteristics (metadata availability, station density, and so on) and to the region examined. So the authors, before every statistical test, should choose and apply a homogenization approach to verify data quality, (see Brunetti et al. 2012 as an example)

Brunetti, M., Caloiero, T., Coscarelli, R., Gullà, G., Nanni, T., Simolo, C. Precipitation variability and change in the Calabria region (Italy) from a high resolution daily dataset, International Journal of Climatology 32 (1), 57-73, 2012.

As regards the trend analysis, why have "not significant" trend values been discussed? A non significant trend either positive or negative has the same meaning as "not different from zero". Recommendation: avoid non-significant trends discussion in the text.

Authors explain they used ordinary kriging to interpolate site data to a grid. They mention that previous studies used multiparametric interpolation including height, sea distance, etc.. Did authors include this information in the presented analysis? Or it was not necessary. I would appreciate some comments regarding this topic.

Authors say that only 4 stations were available in 1914. It's not clear if these data were used for detecting trend. I think variance is very high when only few stations are used to reconstruct a field. Can this affect trend result?

I am not English native speaker but I would appreciate a deep review of language.

Some specific comments in the supplement to this comment.

Please also note the supplement to this comment: http://www.nat-hazards-earth-syst-sci-discuss.net/nhess-2016-330/nhess-2016-330-RC1-supplement.pdf

2016.

СЗ

---

## Referee Comment (RC2) · Anonymous Referee #2 · 4 Jan 2017

**General Comments**

The article deals with the analysis of daily precipitation from AEMET in the Balearic Islands for the period 1950-2009. The authors have compiled an extensive dataset on precipitation observed from the ground in this archipelago and analysed their trends.

The manuscript has a number of merits: It treats a subject of relevance to the audience of NHESS. It is based on the complete dataset of ground precipitation stations of AEMET in the Balearic Islands. Finally, the analyses produce interesting results, as is pointed out in the conclusiond.

However, there are some questions that the authors should solve in a revised version

of the paper before its publication. The main problems are focused on: a) Give proper credit to recent bibliography on the matter b) A poor quality control c) Impact of the orography in the rainfall distribution d) Completeness of lack of data e) Only trends with a statistical significance above 95% or 90% are usually accepted in the trend analysis of precipitation (mainly in these studies potentially related with climate change). Then, you cannot generalise the precipitation decrease and cannot accept as representative those trends with significance inferior to 90%.

The acknowledgements suggest that this PREGRIDBAL database has been obtained by public funding. Consequently, it should be open access (at less from 1950 to 2009). Could you include in the paper the link to this database? It would increase considerably the impact and diffusion of this paper and the authors could require to the users of this database to refer to this paper.

If the dataset used or created by the authors is not made public the scientific contribution of the paper is limited by the fact that other researchers will not be able to verify or question their conclusions, improve on their methods or make further analyses.

The English language should be revised.

In conclusion, in my opinion, this paper deserves publication in NHESS after a few improvements have been made. Please, take into account my previous comments and the following ones.

**Abstract:**

Include the number of stations you have used in the paper for the trend analysis. If you focus the paper on the period 1950 to 2009, why you refer to the previous years? Eliminate these conclusions that are non significant statistically. You speak about weather patterns but they are not explained in the text.

Introduction:

You cite the 4th Assessment Report of the International Panel on Climate Change

(2007). This work has been updated in 2014! Please, refer to the AR5 of IPCC, 2013 (WGI) or 2014 (WGII, WGIII and synthesis).

The Introduction does not give proper credit to the most recent literature on precipitation trends in this region, Spain or the Mediterranean Region, neither to the other daily/monthly databases created in Spain. Consequently, the authors do not consider previous results, and do not include a comparative analysis with them. They speak about the Spain02 daily precipitation database built from 1950 up to 2003 (Herrera et al., 2012), and the trend obtained by Homar et al (2010) for the Balearic Islands. But they do not compare with the trends obtained from this database by other authors for the entire Spain or specific regions near the Balearic Islands. It would be also interesting to present the trends obtained from the MOPREDAS monthly precipitation database (González-Hidalgo et al. 2009 and 2011). These databases, built from AEMET data, have been widely used to explore regional patterns and trends (Vicente-Serrano et al. 2010; Turco and Llasat, 2011). In spite of the different time and/or spatial resolution, it would be necessary to improve the paper and compare with the trends obtained from these databases (or another Mediterranean ones), especially if your results refer to annual values.

You introduce the 1-km resolution grid from Guijarro (1986) over the Balearic Islands for the period 1961-1980, telling that this author have applied the multiparametric interpolation (including height, sea distance, terrain gradient,...). However, it is not clear if you apply this interpolation to this paper

Data and methodology:

Please, include the main physiographic features of the Balearic Islands.

To create a precipitation map of the entire archipelago with only 4 stations can introduce some misunderstandings. Please, indicate from which year you would have a minimum number of stations that could be enough representatives, and start the regional analysis on this year. To which period refers the sentence "Average spatial den-

СЗ

sity of stations on the grid, leaving aside asynchrony in the running of many of them, is 0,08 stations per km2 (or 3,5 km average distance between neighbouring stations)"?

I do not understand the objective of the sentence "thus solving the estimation uncertainty (Dingman et al. 1988): an improved technique after Krige's work (1951) and which has recently advanced drastically in its application in the climatology field and its respective variables (Moral, 2009)". The references of Dingman and Krige are very old, and the krigging application is usual to analyse precipitation fields.

Some aspects of the methodology deserve clearer and more elaborate explanation: ć Which is the criterion used to consider a precipitation day? More than 1 mm/24h? ć The quality control is poor. To eliminate non-physical precipitation values is not enough. Have you made any geographical comparison or correlation between daily values? Perhaps AEMET has applied a preliminary quality control. Do you know it? ć Have you applied any homogeneity criteria to each series? ć How have you complete the gaps? You could fill the gaps in precipitation records through bilinear regression or any other methodology. ć How have you considered the influence of several physiographic characteristics on precipitation? ć Why you define the anomaly as the rate between daily rainfall with respect to average annual precipitation, and not to the monthly one?

You say that you have applied ordinary Kriging for the interpolation, using the daily anomaly with respect to the annual mean for all available observations each day, but this methodology don't work when strong changes in the orography are produced. This is the case of the Tramuntana mountain range. You should consider the physiographic influence of elevation and slopes orientation. One first approach would be the application of the precipitation lapse rate (PLR) that is represented as the ratio between changes in mean annual precipitation and changes in elevation. There is some literature about the influence of the topography and the PLR values like Johansson and Chen (2003), Durán et al (2013), Marquínez et al (2003) or Naoum and Tsanis (2004). Another possibility would be the application of the methodology developed by Guijarro

**(1986)**

Particularly, in the case of Spain, regional linear regression of mean annual precipitation and altitude was assumed in MAGRAMA (2004). In this work, virtual precipitation series at higher altitudes were estimated to improve the interpolation of monthly records by means of inverse distance weighting (IDW) algorithm and an average PLR value was obtained in basis to precipitation series provided by AEMET. Attending to the fact that this work used all the AEMET precipitation series, the Balearic series were also included.

Although your work refers to daily data, some considerations following my previous comments should be considered in the interpolation when you built the PREGRIDBAL database for the Serra de Tramuntana Region.

Paragraph 2.3.1 should be eliminated. This kind of information should be included at the beginning of the section 2, when the AEMET network and the number of stations used should be presented.

The tendency methodology applied is not the usual nowadays and it should be necessary to solve previous questions, like if the null hypothesis in the MannKendall test is satisiňĄed (on the contrary there is more probability of detecting trends when actually none exist). Giving the importance of the results, other methodologies like the "moving block bootstrap" method developed by Kiktev et al. (2003) or other explained in Moberg and Jones (2005) should be applied.

The pattern trend should be analyzed with caution, considering only the maps with iňĄeld signiiňĄcance at least greater than 95%, (i.e. Livezey and Chen, 1983)

Products:

Change the name of this section by Results.

Section 3.2 should be moved to Data and Methodology

All the trends with a significance <90% are not representative and should not be discussed. Please, re-write this section.

Please, avoid specific names of cities that are not showed and marked in the maps.

Attending that you have daily data it would be interesting to analyse the ETCCDI index evolution.

How do you relate the analysis of the two cases of study with the rest of the paper?

Please, if you introduce Figure 5, it should be necessary to introduce any short explanation or reference to justify why this year recorded the maximum annual precipitation

Please, modify some formal aspects of Table 1

Why do you split the period in two: 1950-1979 and 1980-2009? Why do you not use a running window?

Conclusion:

Please, focus only in the results with statistical significance>90%.

What do you mean with "This positive tendency could be attributed to the effect of friction which the city has on atmospheric currents or also to the pollution generated in the urban area"?. Are you mixing turbulence questions with precipitation patterns? How do you can justify it? The pollution level and size of Palma de Mallorca could justify it? In the abstract you speak about changes in the precipitation patterns, but you do not explain anything about it. Which are these changes in precipitation patterns? Have you analysed changes in the frequency of the weather maps (or Principal Components) associated to precipitation?

**References**

Durán L, Sánchez E, Yagüe C. 2013. Climatology of precipitation over the Iberian Central System mountain range. Int. J. Climatol., 33, 2260–2273

González-Hidalgo JC, López-Bustins JA, Štepánek P, Martin-Vide J, de Luis M. 2009. Monthly precipitation trends on the Mediterranean fringe of the Iberian Peninsula during the second-half of the twentieth century (1951–2000). Int. J. Climatol., 29, 1415–1429

González-Hidalgo JC, Brunetti M, de Luis M. 2011. A new tool for monthly precipitation analysis in Spain: MOPREDAS database (monthly precipitation trends December 1945–November 2005). Int. J. Climatol., 31, 715–731

Herrera S, Gutiérrez JM, Ancell R, Pons MR, Frías MD, Fernández J. 2012. Development and analysis of a 50-year high-resolution daily gridded precipitation dataset over Spain (Spain02). Int. J. Climatol., 32, 74–85

IPCC, 2013. Climate Change 2013: The Physical Science Basis, in: Contribution of Working Group I to the Fifth Assessment Report of the Intergovernmental Panel on Climate Change, Cambridge University Press, Cambridge, 1535 pp.

IPCC, 2014, Climate Change 2014: Impacts, Adaptation, and Vulnerability, IPCC Working Group II Contribution to the Fifth Assessment Report of the Intergovernmental panel on climate change http://www.ipcc.ch/report/ar5/wg2/

Johansson B, Chen D. 2003. The influence of wind and topography on precipitation distribution in Sweden: Statistical analysis and modeling. Int. J. Climatol. 23, 1523-1535

Kiktev, D., Sexton, D. M. H., Alexander, L., and Folland, C. K.: Comparison of Modeled and Observed Trends in Indices of Daily Climate Extremes, J. Climate, 16, 3560–3571, 2003.

Livezey, R. E. and Chen, W. Y.: Statistical Field SigniiňAcance and its Determination by Monte Carlo Techniques, Mon. Weather Rev., 111, 46–59, 1983.

Marquínez J, Lastra J, García P. 2003. Estimation Models for Precipitation in Mountainous Regions: the Use of GIS and Multivariate Analysis. J. Hydrol. 270, 1-11

Moberg, A. and Jones, P. D.: Trends in indices for extremes in daily temperature and precipitation in central and western Europe, 1901–99, Int. J. Climatol., 25, 1149–1171, 2005.

Naoum S, Tsanis IK. 2004. Orographic precipitation modeling with multiple linear regression. J. Hydrol. Eng 9(2), 79-102

MAGRAMA 2004. Water in Spain. Ministry of Agriculture, Food and Environment. Technical Secretariat-General. Madrid. Spain

Turco M, Llasat MC. 2011. Trends in indices of daily precipitation extremes in Catalonia (NE Spain), 1951–2003. Nat. Hazards Earth Syst. Sci., 11, 3213–3226

Vicente-Serrano SM, Beguería S, López-Moreno JI, García-Vera MA Stepanek P. 2010. A complete daily precipitation database for northeast Spain: reconstruction, quality control, and homogeneity. Int. J. Climatol., 30: 1146–1163

---

## Author Comment (AC1) · 17 Feb 2017

The comment was uploaded in the form of a supplement:
http://www.nat-hazards-earth-syst-sci-discuss.net/nhess-2016-330/nhess-2016-330-AC1-supplement.pdf
* * *

---

## Author Response (AR1)

Dear Anonymous Reviewer #1:

We appreciate your thoughtful comments on our manuscript, highlighting the relevance and interest of the project we are developing. The product that resulted from Phase I of this mid-term Atlas project is PREGRIDBAL 1.0. This is already expressed in the first two words of the title: "PREGRIDBAL 1.0" and "towards…".

We carefully read your comments and appreciate the suggestions and directions to improve the gridded data in version 2.0. In particular:

**Quality control and homogenization:** The quality of individual measurements we used is overviewed by the official AEMET assistant observers network. These group of trained observers manually read the rain-gauge accumulations on a daily basis and report a "Not available" datum when the quality of the reading is insufficient. As already mentioned in the text (Pag. 4, line 10), these "Not available" records in a series of a particular station are treated as if the station did not exist for that day, just as any other unobserved point across the domain. Given the manual nature of the readings and the heterogeneity of the conditions around the stations, we defined a 5% error to all observed values in the catalog (Page. 5, line 18). This error is transferred throughout the calculations and allow bounding the error associated to the catalog products. Regarding homogeneity problems, caused by modifications in the station surroundings, changes in technology or even changes in the observer's criteria, we acknowledge the lack of a deep homogeneity analysis of the dataset of 418 series of daily accumulations. The homogeneity of highly variable quantities is still a subject of current research, and more specifically the homogeneity of daily rainfall rates. In an area with very high temporal and spatial variability such as the Mediterranean, the challenge is even larger. We plan on applying a homogenization procedure in the preprocess step of the daily values in the next version of the catalog. It is noteworthy that the main source of inhomogeneity in the series from operational networks is the change in location, which is accounted for in the AEMET database, not as a change in location but as the installation of a new station and a cease of operations of the original one. The considerations in Brunetti et al. (2012), together with the algorithms discussed in Guijarro (2014)

Guijarro, José A. 2014. "Quality Control and Homogenization of Climatological Series." CHAP. In *Handbook of Engineering Hydrology: Fundamentals and Applications*, 501–13. CRC Press.

**Trend analysis and significance:** We fully agree with the reviewer comment. We have reread the text, and removed any mention or interpretation to non-significant trend values. This has affected sections 4.1 and 4.2. Since the definition of the pre-specified threshold dividing significant from not significant trends is user dependent, we have avoided the definition of such a unique threshold across the text to allow the reader put his own judgement on the actual significance level of each discussed trend. Nevertheless, only values above 70% are mentioned in the revised text.

**Guijarro (1986) multiparametric analysis:** As described in section 2.3 Analysis, we use an ordinary kriging with precipitation data only, based on a daily exponential variogram fitted to the data available for each day. Future updates of the catalog will certainly consider the inclusion of alternative covariates to improve the final gridded product.

**Period used to detect trend:** Although the daily precipitation maps are available since January 1$^{st}$, 1914, all aggregate and trend products are produced using data from January 1$^{st}$ 1950. We modified the text (Pag. 3, line 29 and Pag. 5 line 23) to make this clearer to the reader.

English has been revised and improved across the document. We appreciate the comment and understand the concern also as not native English speakers.

We appreciate the time and effort put by the reviewer highlighting specific comments on the pdf. We have addressed all these comments in the revised version.

Dear Anonymous Reviewer #2:

We appreciate your thoughtful comments on our manuscript, highlighting the merits, relevance and interest of the project we are developing. The product that resulted from Phase I of this mid-term Atlas project is PREGRIDBAL 1.0.

We carefully read your comments and appreciate the suggestions and directions to improve the gridded data in version 2.0. In particular:

Recent bibliography:

**Quality control:** The quality of individual measurements we used is overviewed by the official AEMET assistant observers network. These group of trained observers manually read the rain-gauge accumulations on a daily basis and report a "Not available" datum when the quality of the reading is insufficient. As already mentioned in the text (Pag. 4, line 10), these "Not available" records in a series of a particular station are treated as if the station did not exist for that day, just as any other unobserved point across the domain. Given the manual nature of the readings and the heterogeneity of the conditions around the stations, we defined a 5% error to all observed values in the catalog (Page. 5, line 18). This error is transferred throughout the calculations and allow bounding the error associated to the catalog products.

**Use of covariant parameters (e.g. orography):** As described in section 2.3 Analysis, we use an ordinary kriging with precipitation data only. Future updates of the catalog will consider covariates such as the orography or the distance to the sea to improve the final gridded product. We are confident that these updates will lead to significant improvements in the final products. This is left for future work.

**Significance of precipitation trends:** Although we agree that it is common practice to define a hard threshold to divide significant from not significant trends, the interpretation of the statistical significance as the probability of detecting an actual change in the underlying process converts this artificial threshold in a subjective mark. Nonetheless, we have modified the text and eliminated all mentions and discussions referring to trends with significances below the 70% level. We avoid mentioning the *significant/not significant* character of the trends but rather inform the reader about the specific confidence on that trend. This allows the readers to make informed decisions about our results.

**Access to database:** As much as we agree with the open-accessibility of data collected using public funds, we are unfortunately bind by a MoU with AEMET that does not allow us to share this data. However, all maps and spatial accumulation graphs are made available through the open web portal http://pregridbal-v1.uib.es.

English has been revised and improved across the document. We appreciate the comment and understand the concern also as not native English speakers.

We appreciate the time and effort put by the reviewer highlighting specific comments on the pdf. We have addressed all these comments in the revised version.

**ABSTRACT**
**Number of stations in abstract:** done. Thanks.
**Reference to previous years:** the catalog that this article presents includes rudimentary daily precipitation maps for the first half of the XX century, which admittedly have limited decision-making value but are left in the catalog with all precautionary notes for completeness and historical reference.

**"weather pattern":** This must be a misunderstanding since no reference to weather pattern can be found in the text. The expression "precipitation patterns" is used twice as a generic description of precipitation distribution across the territory.

**INTRODUCTION**

**Reference to 5[th] assessment:** done. Thanks.

**Credit to recent literature:** We appreciate the aim of the reviewer to make the connections of our paper more explicit in the text. We have referenced the works regarding the MOPREDAS in the introduction as interesting monthly precipitation analysis over continental Spain, but cannot follow the reviewer's suggestion of performing a comparison with our results for obvious geographical reasons.

**DATA AND METHODOLOGY**

**Brief description of Balearic Islands physiography:** done. Thank you.

**Spatial density of stations:** The key in this sentence is the expression "leaving asynchrony aside", which expresses that this calculation is done using the location of all sites available in the database. In any case, this calculated density is a good estimate of the actual density from 1950 onwards. Krige and Dingman references: These are two well-known references that initiated the geostatistical analysis and the objective of using them is to acknowledge their contribution.

**Precipitation day:** we proceeded to apply the methodology over days with at least one measurement of 24h accumulation of 0.1 mm or more. We don't believe this is noteworthy to mention it in the text.

**Homogeneity criteria:** About homogeneity problems, caused by modifications in the station surroundings, changes in technology or even changes in the observer's criteria, we acknowledge the lack of a deep homogeneity analysis of the dataset of 418 series of daily accumulations. The homogeneity of highly variable quantities is still a subject of current research, and more specifically the homogeneity of daily rainfall rates. In an area with very high temporal and spatial variability such as the Mediterranean, the challenge is even larger. We plan on applying a homogenization procedure in the preprocess step of the daily values in the next version of the catalog. It is noteworthy that the main source of inhomogeneity in the series from operational networks is the change in location, which is accounted for in the AEMET database, not as a change in location but as the installation of a new station and a cease of operations of the original one. The considerations in Brunetti et al. (2012), together with the algorithms discussed in Guijarro (2014)

**Anomaly with respect to monthly mean:** This is an interesting suggestion that we will consider in future updates of the catalog. Thank you.

**"this methodology don't work when strong changes in the orography are produced":** This is actually false. Anomalies are defined locally at the station level, with respect to local annual averages. No geographical transference of information occur when defining the anomalies. This is described in the modified section 2.3 of the text.

**Relocation of section 2.3.1:** We agree with the reviewer suggestion. The paragraph describing the error characteristics of the raingauges has been moved as suggested.

**RESULTS**

**Name of the section "Results" (previous "Products"):** We fully agree with this comment. Done.

**Section 3.2:** This sub-section presents results, and so we think it should not be moved to the "Data and Methodology" section.

**ETCCDI index:** We appreciate this comment and will consider this diagnostics in future updates of the catalog.

**Case studies:** These are clearly presented as illustrative examples of the daily precipitation maps. We strongly believe they help illustrate the type of daily products we obtain.

**Physical causes of maximum annual precipitation (Fig. 5)**: The explanation of the causes for this year to be the record is subject of a new research, so it is beyond the scope of this paper. The objective of showing this map is merely illustrative to help the reader realize the products available in the catalog.

**CONCLUSIONS**

**"Friction and pollution":** Please note that this sentence is expressed with a degree of uncertainty but the mentioned mechanisms contribute in the long term to the increase of precipitation by increasing the number of condensate nuclei and slightly contributing to the stagnancy of precipitating systems. We believe this ideas are worth mentioning provided the speculative tone of the phrasing.

---

## Author Response (AR2)

REF: **nhess-2016-330**

**PREGRIDBAL 1.0: towards a high-resolution rainfall atlas for the Balearic Islands (1950-2009)**
Toni López Mayol, Víctor Homar, Climent Ramis, and José Antonio Guijarro

Dear Editor,

We appreciate your suport in the editorial processing of this manuscript, which we strongly believe contains interesting information for the NHESS reader. We also acknowledge the interesting suggestions made by the reviewers which have helped improve the quality and readability of the final version.

We have gone through the document and made numerous language modifications to make the text clearer.

Best regards,

Victor Homar and coauthors.